# α-Fe_2_O_3_/, Co_3_O_4_/, and CoFe_2_O_4_/MWCNTs/Ionic Liquid Nanocomposites as High-Performance Electrocatalysts for the Electrocatalytic Hydrogen Evolution Reaction in a Neutral Medium

**DOI:** 10.3390/ijms25137043

**Published:** 2024-06-27

**Authors:** José Ibarra, María Jesus Aguirre, Rodrigo del Río, Rodrigo Henriquez, Ricardo Faccio, Enrique A. Dalchiele, Roxana Arce, Galo Ramírez

**Affiliations:** 1Departamento de Química Inorgánica, Facultad de Química, Pontificia Universidad Católica de Chile, Av. Vicuña Mackenna 4860, Casilla 306, Correo 22, Santiago 8331150, Chile; jfibarra@uc.cl (J.I.); rdelrioq@uc.cl (R.d.R.); 2Millennium Institute on Green Ammonia as Energy Vector (MIGA), Av. Vicuña Mackenna 4860, Macul, Santiago 7820436, Chile; maria.aguirre@usach.cl; 3Departamento Química de los Materiales, Facultad de Química y Biologia, Universidad de Santiago de Chile, Av. B O’Higgins 3363, Estación Central, Santiago 9170022, Chile; 4Instituto de Química, Facultad de Ciencias, Pontificia Universidad Católica de Valparaíso, Av. Brasil 2950, Valparaíso 2362807, Chile; rodrigo.henriquez@pucv.cl; 5Área Física & Centro NanoMat, DETEMA, Facultad de Química, Universidad de la República, Av. Gral. Flores 2124, CC 1157, Montevideo 11800, Uruguay; rfaccio@fq.edu.uy; 6Instituto de Física, Facultad de Ingeniería, Universidad de la República, Herrera y Reissig 565, C.C. 30, Montevideo 11000, Uruguay; dalchiel@hotmail.it; 7Departamento de Ciencias Químicas, Facultad de Ciencias Exactas, Universidad Andrés Bello, Av. República 275, Santiago 8370146, Chile

**Keywords:** metal oxide/carbon nanotubes/ionic liquid electrode, H_2_ generation, HER catalysis, cobalt ferrite/carbon nanotubes/ionic liquid electrode

## Abstract

Transition metal oxides are a great alternative to less expensive hydrogen evolution reaction (HER) catalysts. However, the lack of conductivity of these materials requires a conductor material to support them and improve the activity toward HER. On the other hand, carbon paste electrodes result in a versatile and cheap electrode with good activity and conductivity in electrocatalytic hydrogen production, especially when the carbonaceous material is agglomerated with ionic liquids. In the present work, an electrode composed of multi-walled carbon nanotubes (MWCNTs) and cobalt ferrite oxide (CoFe_2_O_4_) was prepared. These compounds were included on an electrode agglomerated with the ionic liquid N-octylpyridinium hexafluorophosphate (IL) to obtain the modified CoFe_2_O_4_/MWCNTs/IL nanocomposite electrode. To evaluate the behavior of each metal of the bimetallic oxide, this compound was compared to the behavior of MWCNTs/IL where a single monometallic iron or cobalt oxides were included (i.e., α-Fe_2_O_3_/MWCNTs/IL and Co_3_O_4_/MWCNTs/IL). The synthesis of the oxides has been characterized by X-ray diffraction (XRD), RAMAN spectroscopy, and field emission scanning electronic microscopy (FE-SEM), corroborating the nanometric character and the structure of the compounds. The CoFe_2_O_4_/MWCNTs/IL nanocomposite system presents excellent electrocatalytic activity toward HER with an onset potential of −270 mV vs. RHE, evidencing an increase in activity compared to monometallic oxides and exhibiting onset potentials of −530 mV and −540 mV for α-Fe_2_O_3_/MWCNTs/IL and Co_3_O_4_/MWCNTs/IL, respectively. Finally, the system studied presents excellent stability during the 5 h of electrolysis, producing 132 μmol cm^−2^ h^−1^ of hydrogen gas.

## 1. Introduction

Water electrolysis is one of the most efficient and environmentally friendly methods for producing molecular hydrogen. This gas can be used directly as fuel or used in power generation devices such as fuel cells, making hydrogen one of the leading candidates for decarbonizing global energy demand [1,2,3]. Hydrogen evolution reaction (HER) is vital for sustainable hydrogen production. In this context, electrolysis in neutral media presents a compelling approach with unique advantages. Notably, neutral electrolytes minimize the corrosion of electrodes and enhance material compatibility, addressing challenges in acidic or alkaline environments. This expanded compatibility broadens the range of suitable electrocatalysts, contributing to cost reduction and promoting wider options for green hydrogen production [4]. However, the big obstacle to large-scale hydrogen generation lies in the electrocatalysts used. Traditional compounds used in electrolyzers, such as platinum and iridium, are electrocatalysts with great activity and efficiency toward HER and then hydrogen gas production; however, they are scarce materials, raising the price of industrial electrolyzers [5,6]. In response to this challenge, researchers are actively exploring materials design strategies to reduce production costs without compromising the activity and efficiency of hydrogen generation. One of the systems studied is the carbon paste electrodes [7,8,9]; these paste electrodes are made up of nanometric carbonaceous compounds and a binder agent, which offer numerous advantages in electrocatalysis due to their easy modification, renewable surface, low background current, wide potential window, low cost, and simple fabrication [7]. Among the different carbonaceous pastes, the pastes formed with carbon nanotubes and agglutinated with ionic liquids improve the activity of different electrochemical reactions due to several synergistic effects [10]. The ionic qualities of the ionic liquids, such as their high ionic conductivity and electrochemical stability, increase the electrode’s electrochemical activity and conductivity compared to traditional binders like mineral oil. This enhancement arises from the ionic liquids’ ability to facilitate charge transfer at the electrode–electrolyte interface, improving the kinetics of the electrochemical reactions. Additionally, the ionic liquid’s ability to disperse and stabilize carbon nanotubes within the paste leads to a more uniform and conductive electrode surface, further boosting its performance [10,11,12]. In this context, electrodes formed with multi-walled carbon nanotubes (MWCNTs) agglutinated with the ionic liquid N-octylpyridinium hexafluorophosphate (MWCNT/IL) showed good activity toward hydrogen production [13]. This particular combination leverages the unique properties of both MWCNTs, with their high surface area and electrical conductivity, and the ionic liquid, with its charge transport capabilities, the promotion of proton adsorption, and the hydrophobic nature of the cation [14,15]. The resulting MWCNT/IL electrode exhibits enhanced electrocatalytic activity for the HER, and further investigations into optimizing the composition and structure of these electrodes could lead to even greater improvements in their performance for hydrogen production and other electrochemical processes. In this context, low-cost materials with great abundance and access have been investigated [16,17,18]. Transition metal compounds have been studied in different electrochemical reactions, in particular those with energetic interest (oxygen reduction reaction (ORR), hydrogen evolution reaction, and oxygen evolution reaction (OER)) showing great activity and interesting uses [18,19]. In the case of HER, transition metal oxides such as cobalt, iron, nickel, zinc, etc., have been studied [20,21]. Those metal oxides, especially the spinel ferrite family (MFe_2_O_4_), have emerged as a promising electrocatalyst for hydrogen evolution reactions due to several key advantages. Its earth-abundant and cost-effective composition displays excellent potential as an electrocatalyst for the hydrogen evolution reaction. CoFe_2_O_4_ is particularly effective at catalyzing the HER due to its mixed-valence nature and flexible crystal structure, enabling efficient charge transfer and adsorption of reactive substances. It is also highly stable across various pH conditions, resistant to corrosion, and durable over time, ensuring long-term operational feasibility. One of its key benefits is that its properties can be adjusted by changing the synthesis conditions, making it possible to optimize its performance for specific HER requirements [22,23,24,25]. However, to improve the activity toward the hydrogen evolution reaction, transition metal semiconductors require electron-rich environments or delocalized systems to enhance their conductivity and electronic transfer, which decreases, especially in alkaline and neutral media [26]. With this background, this work proposes a carbon paste electrode made with MWCNTs agglutinated with the ionic liquid N-octylpyridinium hexafluorophosphate where cobalt ferrite (CoFe_2_O_4_) is occluded, forming the CoFe_2_O_4_/MWCNTs/IL electrocatalyst. This electrode was studied toward hydrogen evolution reaction in a neutral medium and compared with similar transition metal oxides. The obtained activity by the occlusion of the transition metal oxide in the MWCNT/IL electrocatalyst results in a promising material for HER with excellent stability and durability.

## 2. Results and Discussion

### 2.1. Structural Characterization

The synthesis of the ionic liquid N-octylpyridinium hexafluorophosphate (IL) (see Figure 1) could be corroborated by the ^1^H-NMR study, where the H spectrum of the organic cation of the molten salt was analyzed. The shifts of the peaks obtained are presented in Table 1. These signals agree with the spectrum previously reported in the literature [8], corroborating the correct synthesis of the ionic liquid.

To ascertain the structural characteristics, X-ray diffraction (XRD) has been used to characterize the single MWCNTs and the prepared metal oxides/MWCNTs/IL nanocomposite samples. From Figure 2a it can be seen that the XRD pattern of the MWCNTs sample shows the typical strong diffraction peak at 2θ = 26.2° corresponding to the (002) reflection of the hexagonal graphite structure (JCPDS: # 96-101-1061). The diffraction peaks at around 43° and 45° are due to (100) and (101) graphitic planes, respectively (JCPDS: # 96-101-1061) [27]. Figure 2b–d show the XRD patterns for the different prepared transition metal oxides/MWCNTs/IL nanocomposite samples. In all of these three XRD patterns, the presence of a diffraction peak at 2θ angle of ~26° can be seen, which can be assigned to the MWCNTs phase, according to the XRD of this material (see Figure 2a). Moreover, it can be seen that the diffraction peaks corresponding to the cobalt and iron oxides agree very well with those reported in the literature (JCPDS 85-0599 and JCPDS 78-1970, respectively) [28], which are related to the compound Co_3_O_4_, with a cubic crystalline structure and to hematite (α-Fe_2_O_3_), with a hexagonal crystalline structure (see Figure 2b,c). In the case of cobalt oxide, this compound exhibits a crystallite size of 38 nm and a crystal lattice parameter of a = 8.059 Å, parameters that coincide with literature (JCPDS 85-0599) [28]. In the case of iron oxide, with a lattice parameter of a = 5.008 Å and c = 13.63 Å, the formation of the hematite phase is corroborated (JCPDS 78-1970). This oxide also has a nanometric structure exhibiting a crystallite size of 27 nm. The crystallite size values obtained through the Scherrer equation and the crystal lattice parameters for the different studied compounds are depicted in Table 2. Furthermore, the crystalline nature of the CoFe_2_O_4_/MWCNTs/IL nanocomposite sample has also been characterized by XRD. As it can be appreciated in Figure 2d, the discernible diffraction peaks can be indexed to (220), (311), (222), (331), and (422)/(333) planes of a cubic structure CoFe_2_O_4_, which are characteristics for CoFe_2_O_4_ in cubic spinel-type structure and match well with the standard data of CoFe_2_O_4_ (JCPDS no. 77-0426) [29]. The peaks with the question marker probably correspond to impurities of cobalt oxide phases (400) and (440), respectively. On the other hand, to corroborate the obtention of the metal oxides, XRD and FT-IR of the compounds without the carbonaceous paste were performed (see Appendix A). Finally, by using Scherrer’s equation, it could be ascertained that the broadened diffractive peaks of CoFe_2_O_4_ indicate that the crystalline size of CoFe_2_O_4_ particles is 34 nm (see Table 2).

Raman spectroscopy shows the signals corresponding to the carbon nanotubes in the bimetallic oxide system and their respective MWCNTs and MWCNTs/IL blanks (see Figure 3). The observed signals allow us to obtain the nanotubes’ ordering coefficient (I_D_/I_G_) in the absence and presence of the bimetallic oxide. I_D_/I_G_ is obtained from the intensities of the D and G bands of the carbonaceous material, and the presence of sp^2^ and sp^3^ hybridizations is considered. Thus, the higher I_D_/I_G_, the greater the disorder due to a more significant presence of defects in the material studied. Therefore, it can be seen in Table 3 that, when modifying the carbonaceous compound, either through agglutination with the ionic liquid or with the inclusion of the iron-cobalt oxide, an ordering of the carbonaceous structure is produced (lower Id/Ig). On the other hand, the relationship between the 2D and D bands can be obtained from the I_2D_/I_G_ ratio. This coefficient represents the long-distance structural order [32], where the higher the value, the higher the order, and it can be seen that by adding cobalt iron oxide, an increase in this factor occurs, generating a more significant long-distance order in the structural system. As mentioned before, the addition of the ionic liquid and cobalt ferrite does not significantly alter the overall defect and graphitic nature of the MWCNTs; the improvement in crystallinity could facilitate more efficient charge transfer along the MWCNTs, thereby enhancing their electrical conductivity and promoting faster electron kinetics during the HER process [33].

### 2.2. Morphological and Chemical Composition Study

The morphology of the synthesized oxides and the prepared carbonaceous pastes are seen in Figure 4. The FE-SEM images A, C, and E correspond to iron, cobalt, and cobalt ferrite oxides, respectively. It is observed that the synthesized oxides have a nanometric size, with an average particle size of 97 nm for iron oxide, 68 nm for cobalt oxide, and 122 nm for cobalt ferrite. Each oxide has relatively homogeneous nanoparticle sizes. On the other hand, images B, D, and F show the systems formed by the Co_3_O_4_/MWCNTs/IL, Fe_2_O_3_/MWCNTs/IL, and CoFe_2_O_4_/MWCNTs/IL pastes, respectively. The carbonaceous paste agglutinated with the ionic liquid generates a robust matrix where the nanoparticles are dispersed homogeneously. Carbon nanotubes and the different oxides (brightest elements in the image) are clearly observed. EDX was performed on the prepared composites (Figure 5), where the elements C, P, and F are observed, corresponding to the carbonaceous paste agglutinated with the ionic liquid. The elements Fe and O correspond to the presence of the oxides in the hematite; Co and O correspond to the cobalt oxide, and both metals in the sample containing the cobalt ferrite are also observed. The presence of chlorine and sodium is also observed, which would be impurities from the reactants used in the synthesis [34].

### 2.3. Electrochemical HER Activity and Stability of the Different Studied Transition Metal Oxide Nanocomposite Electrocatalysts

The electrochemical studies were carried out in a neutral medium, using a pH 7 phosphate buffer as a supporting electrolyte. The nanocarbon pastes show increased catalytic activity when including the different nano oxides compared to the MWCNTs/IL electrode. The CoFe_2_O_4_/MWCNTs/IL system shows a decrease in the applied potential to produce the hydrogen discharge (see Figure 6A). Thus, a discharge is observed at approximately −270 mV vs. RHE, which, in comparison with the system with cobalt oxide (−540 mV) and iron oxide (−530 mV), corroborates the increase in activity by the system composed of cobalt ferrite. Additionally, the system was compared with the well-known platinum catalyst, which presents an onset potential of −170 mV. The obtained activity is also compared with similar systems studied for HER. Medang et al. proposed a green synthesized cobalt ferrite, which was deposited in GC, obtaining 440 mV of overpotential in alkaline media [35]. Cobalt and nickel ferrite-based graphene was proposed by Nivetha et al., reaching an onset potential of −0.98 V in acidic media [23]. This comparison underscores the significance of the −270 mV onset potential achieved with the cobalt ferrite carbon paste, suggesting its potential for improved hydrogen evolution processes. On the other hand, the current at an applied potential of 300 mV shows an increase, probably due to the adsorption of protons on the electrode surface, which would correspond to the beginning of the reaction at this point of the linear scan [36]. The Tafel slopes at low applied potentials are similar in all systems (see Figure 6C), which would be due to the proton adsorption mechanism mentioned above. According to the Tafel slope values reported in the literature [37], the mechanism would be limited by the adsorption of two protons in nearby active sites for the subsequent formation of molecular hydrogen from the transfer of electrons (Volmer–Heyrovsky mechanism). The reason for the values over the typically 120 mv/dec, (see Table 4) could be that in electrocatalytic water splitting for H_2_ production in neutral or alkaline media, HER kinetics are slowed by the requirement of an additional water dissociation step, making the electrocatalyst need extra energy to gain a proton for H_2_ evolution [4,38]. Electrochemical stability, that is, the ability of the electrocatalyst to maintain the initial electrocatalytic activity for a long time, is an important parameter to evaluate whether the electrocatalyst is suitable for industrialized applications [39]. Figure 6B shows that the systems studied are electrochemically stable over time at a fixed potential E = −890 mV. It was measured during two hours of electrolysis, and a drop in the current density was not observed in the studied systems, thus corroborating the robustness of the nanocarbonaceous pastes (MWCNTs/IL) linked to the nanostructured transition metal oxides. On the other hand, the CoFe_2_O_4_/MWCNTs/IL system even presents a gradual increase in current density. This behavior can be attributed to the increment in the effective area of the CoFe_2_O_4_/MWCNTs/IL phase in contact with the electrolytic solution according to immersion time, which accounts for, in turn, an increase in the electroactivity of the bimetallic oxide compound.

The electrochemical performance and the electrochemical events occurring within the electrocatalyst interface were further investigated using electrochemical impedance spectroscopy (EIS) analysis. Figure 6D shows the Nyquist diagrams of the studied systems obtained from EIS, in which it can be corroborated that the CoFe_2_O_4_/MWCNTs/IL system corresponds to the electrode with the lowest charge transfer resistance, inferred from the smaller diameter of the semicircle [40]. The experimental data have been fitted by considering a conventional Randless circuit, and the values of solution resistance (Rs), charge transfer resistance (Rct), and the value of the constant phase element (CPE-P) were calculated (Table 5). The elevated solution resistance (Rs) values observed are likely attributable to the consequences of part of the electrode bound with mineral oil, which electrically insulates these surfaces. This is particularly evident in the Co_3_O_4_/MWCNTs/IL composite, which added to the inherent low conductivity of the cobalt oxide, presenting a larger Rs value than the other electrodes. This behavior can be confirmed in the slightly resistive profile observed in the linear sweep voltammetry of this composite.

### 2.4. Evolved Hydrogen Quantification through Gas Chromatography

The evolved hydrogen gas was quantified using gas chromatography to corroborate the electroactivity of the different transition metal oxide nanocomposite electrocatalysts in terms of hydrogen yield production. To do this, electrolysis was performed (Figure 7) in an appropriately sealed electrochemical cell with a 5 mL headspace. After 5 h of electrolysis, the gas produced in the headspace has been extracted into a 50 μL syringe and measured in the gas chromatograph.

The signal obtained in the gas chromatograph is analyzed using a previously prepared calibration curve (see Appendix A). The obtained chromatographic signal allowed us to quantify the produced hydrogen gas given for the case depicted in Figure 7A. It can be seen that, as was mentioned before, the current density increases during the electrolysis product with an increase in the effective area of the electrode, in this case reaching up to a plateau around −12 mA/cm^2^. This fact is in agreement with the CPE-P value obtained by simulation of the EIS responses to CoFe_2_O_4_/MWCNTs/IL shown in Table 5. Considering a geometric electrode area of 0.03 cm^2^, an electrolysis time of 5 h, the 50 μL volume of the syringe, and extrapolating into the headspace of the cell, it is calculated that the CoFe_2_O_4_/MWCNTs/IL system releases 132 μmoles h^−1^ cm^−2^. Faraday’s law determined the Faradaic efficiency of hydrogen production (Figure 7B), where the amount of quantified hydrogen gas (19.8 μmoles) corresponds to 68.8% of the moles theoretically calculated from the transferred electronic charge. While these results indicate that the CoFe_2_O_4_/MWCNTs/IL electrocatalyst is efficient and stable toward the HER, with higher and more efficient hydrogen production than that reported in our previous work (see Table 6) [13], it is important to address the remaining 31.2% of the current not accounted for by hydrogen generation. We hypothesize that a significant portion of this unaccounted current is attributed to the oxygen reduction reaction (ORR). Given the use of an aqueous electrolyte and the open cell configuration of our experimental setup, it is plausible that oxygen generated at the counter electrode during water electrolysis could diffuse into the bulk solution and undergo reduction at the working electrode. Further investigation involving the use of a divided cell setup is necessary to confirm this hypothesis and precisely quantify the contribution of the ORR to the overall current distribution.

## 3. Materials and Methods

### 3.1. Reagents and Equipment

All reagents were of analytical grade, purchased by Sigma-Aldrich Chile, and used without any previous treatment. Ultrapure water was obtained from a Heal Force Smart-N purification system with 18.2 MΩ cm. Argon and oxygen gases were obtained by Lynde (Chile) with a 99.9% purity. Structural characterization of the samples has been examined by X-ray powder diffraction (XRD) measurements, carried out with standard theta-2 theta scans on a Philips PW180 diffractometer (30 kV, 40 mA, CuKα radiation with λ = 1.5406 Å). Raman spectroscopy was carried out using an excitation wavelength of 532 nm on Alpha 300-RA Confocal Raman Microscopy equipment, WITec, Ulm, Germany. A field emission scanning electron microscope (FE-SEM with energy dispersive X-ray spectroscopy (EDX)) QUANTA FEG250 Thermo Fisher Scientific Co., Ltd., Hillsboro, OR, USA) was used to determine the morphology and composition of the different synthesized samples. These samples were supported on glassy carbon plates and analyzed before the electrochemical studies.

### 3.2. Ionic Liquid Synthesis

The synthesis of the ionic liquid was divided into two parts. Firstly, the N-octylpyridinium bromide ionic liquid was synthesized by adding 10 mL of pyridine, 21.5 mL of bromooctane, and 20 mL of acetonitrile to a synthesis flask. The reaction was carried out by reflux, under an argon atmosphere, with constant stirring for 48 h at 100 °C. Once the reaction concluded, the solvent was evaporated, and the final product was dried at 70 °C for 15 h [46]. Then, the ion exchange was carried out by mixing 12.54 g of N-octylpyridinium bromide and 8.70 g of potassium hexafluorophosphate in a synthesis flask. Then, 20 mL of water and 20 mL of dichloromethane were added and stirred at room temperature for two hours. The resulting solution was transferred from the flask to a separating funnel and washed with abundant water, saving the organic phase. Silver nitrate was added to the remaining aqueous phase in the separating funnel to quantify the bromide ion by precipitation. When there is no precipitate, the washing process is finished. Finally, the solvent of the organic phase was evaporated, resulting in a white serous solid, which was further purified by recrystallization in 95% ethanol.

### 3.3. Cobalt Oxide Synthesis

CoCl_2_ and NaHCO_3_ were weighed in the same molar proportion (1:1) and dissolved in distilled water forming two solutions. The cobalt solution was magnetically stirred for 15 min. Then, the sodium bicarbonate solution was added dropwise and with constant stirring in the cobalt chloride solution. The solution was kept in an ice bath during the whole process. After 15 min of reaction, the product was collected by centrifugation and washed adequately with distilled water. Finally, the solid product was dried in an oven at 100 °C and calcined at 600 °C for 2 h in a tube furnace to produce uniform cobalt oxide nanoparticles with a clear crystalline structure [47].

### 3.4. Iron Oxide Synthesis

Iron oxide (α-Fe_2_O_3_) was synthesized using a coprecipitation method, where 10 g of FeCl_3_ ∗ 6 H_2_O was dissolved in 50 mL of distilled water. This solution was kept at room temperature under constant stirring. Then, 2 mL of a 25% *m*/*m* NH_4_OH solution was added drop by drop at 1 mL per minute. The resulting solution was stirred for one hour, and the pH of the solution remained constant (pH = 1) throughout the precipitation process. Finally, the water from the solution was evaporated by heating at 100 °C for 2 h, obtaining a brown powder, which was further annealed at 500 °C in a tubular oven for four hours in an air atmosphere [48].

### 3.5. Cobalt Ferrite Synthesis

For cobalt ferrite synthesis, 25 mL of 0.4 M iron chloride solution and 25 mL of 0.2 M cobalt chloride solutions were mixed in ultrapure water; 25 mL of 3 M sodium hydroxide solution was prepared and slowly added to the salt solution dropwise. The pH of the solution was constantly monitored as the NaOH solution was added until a pH level of 11–12 was reached under constant stirring.

The liquid precipitate was then heated to a temperature of 80 °C and stirred for one hour. The product was then cooled to room temperature. To obtain particles free of sodium and chlorine, the precipitate was washed twice with ultrapure water and then with ethanol. The beaker’s contents were centrifuged for 15 min at 3000 rpm to separate the supernatant liquid. The supernatant liquid was removed, and the solution was centrifuged until only a thick black precipitate remained. The precipitate was then dried for 12 h at 100 °C. Finally, the acquired substance was ground to obtain a fine powder. At this stage, the product (CoFe_2_O_4_) contains some associated water (up to 10 wt%), which was removed by heating it at 600 °C for 10 h [49].

### 3.6. Electrode Preparation/Modification and Electrochemical Characterization

The transition metal oxides/MWCNTs/ionic liquid nanocomposite electrodes were first prepared by mixing 0.035 g of MWCNTs, 0.02 g of ionic liquid, 72 μL of mineral oil, and 0.035 g of the transition metal oxide in question (i.e., Co_3_O_4_, α-Fe_2_O_3_, and CoFe_2_O_4_), in an agate mortar. Then, small portions of diethyl ether are added, and the nanomaterial content is mixed in the agate mortar until the diethyl ether is evaporated. The nanocomposite resulting paste was introduced in a hollow Teflon electrode with a geometric area of 0.03 cm^2^. Finally, the electrode is heated at 60 °C to achieve a good ionic liquid dispersion and to evaporate the remaining diethyl ether. In detail, the following electrocatalyst electrodes have been prepared: MWCNTs/IL, Co_3_O_4_/MWCNTs/IL, α-Fe_2_O_3_/MWCNTs/IL, and CoFe_2_O_4_/MWCNTs/IL. Electrochemical experiments were conducted with a CH-Instrument CHI750D (CH Instruments, Austin, TX, USA) potentiostat. A conventional single-compartment three-electrode cell has been used, employing the synthesized/prepared composite electrocatalyst, an Ag/AgCl (3 M KCl), and a platinum wire as a working, reference, and counter electrodes, respectively. The electrolyte used was an aqueous 0.1 M phosphate (K_2_HPO_4_/KH_2_PO_4_) buffered saline (PBS) solution with pH = 7.0. In the present work, the potentials were, however, referenced with respect to a reversible hydrogen electrode (RHE) using the following equation:*E*_RHE_ = *E*_Ag/AgCl_ + 0.0591 pH + 0.1976 V(1)

The electrolyte solution was purged with argon for 30 min before each experimental series and then kept under flowing argon out of the solution during the electrochemical analysis. Before any electrochemical measurement, the electrodes have been stabilized by ten cyclic voltammetry (CV) cycles from 0 mV to −1500 mV at 100 mV/s in the PBS electrolytic bath solution, pH = 7. HER activity of the nanocomposite electrocatalyst electrodes was evaluated by linear scan voltammetry (LSV) at a scan rate of 100 mV/s in the PBS electrolytic bath solution, pH = 7, at room temperature. Electrochemical impedance experiments were carried out in a CH-Instrument CHI750D (CH Instruments, Austin, TX, USA) electrochemical workstation in the frequency range between 1 Hz and 100 kHz using a single sinusoidal AC potential of small amplitude (5 mV) to avoid nonlinearity. Experimental impedance data were analyzed and fitted using ZView software 4.0i (Scribner Associates). The electrochemical HER test was initiated, and the evolved hydrogen gas was analyzed by gas chromatography (GC-2030 Plus, Shimadzu, Japan) with a thermal conductivity detector (TCD). Argon was used as the carrier gas. The carbon paste electrodes were prepared by mixing 0.035 g of carbon nanotubes, 0.02 g of ionic liquid, 72 µL of mineral oil, and 0.035 g of the transition metal oxide in an agate mortar. Then, small portions of diethyl ether are added, and a homogeneous mixture is mixed and formed until the solvent evaporates. The carbon paste was introduced in a hollow Teflon electrode with a geometric area of 0.03 cm^2^. Finally, the electrode is heated at 60 °C to the correct dispersion of the ionic liquid and to evaporate the remaining diethyl ether. Before any electrochemical measure, the electrode is stabilized with ten cycles in cyclic voltammetry using phosphate buffer pH 7. The following pastes were prepared: MWCNTs/IL, Co_3_O_4_/MWCNTs/IL, a-Fe_2_O_3_/MWCNTs/IL, and CoFe_2_O_4_/MWCNTs/IL.

## 4. Conclusions

In summary, it could be corroborated by XRD, FE-SEM, and EDX the nanometric structure and chemical composition of the transition metal oxides, as well as the carbon paste with those compounds occluded. Raman spectroscopy shows that the occlusion of the materials studied does not considerably affect the structure and activity of MWCNTs. The electrochemical studies show that carbon paste/cobalt ferrite electrodes improve the activity and stability of hydrogen evolution reaction compared with iron and cobalt oxide paste electrodes; this is probably due to the interaction of both cations (Co^2+^, Fe^3+^) in the same structure providing a better charge transference, added to the great conductivity and high availability of actives sites of the porous composite. This is also corroborated by EIS and the Tafel slopes study. Finally, the system (CoFe_2_O_4_/MWCNTs/IL) is studied toward hydrogen evolution reaction at pH 7, showing excellent stability with a hydrogen production rate of 132 mol/cm^2^ h, resulting in a great alternative as a catalyst for HER in neutral media.

## Figures and Tables

**Figure 1 ijms-25-07043-f001:**
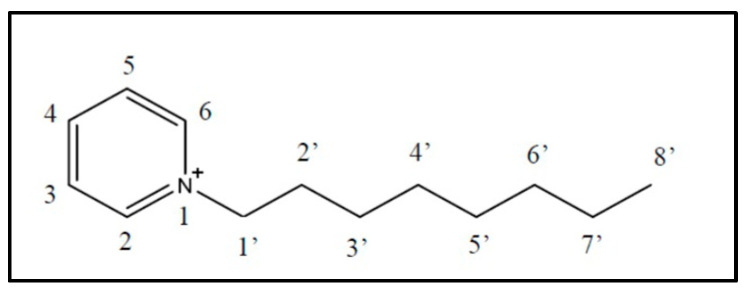
Organic cation of ionic liquid N-octylpiridinium hexafluorophosphate. Numbers in each proton are related to Table 1.

**Figure 2 ijms-25-07043-f002:**
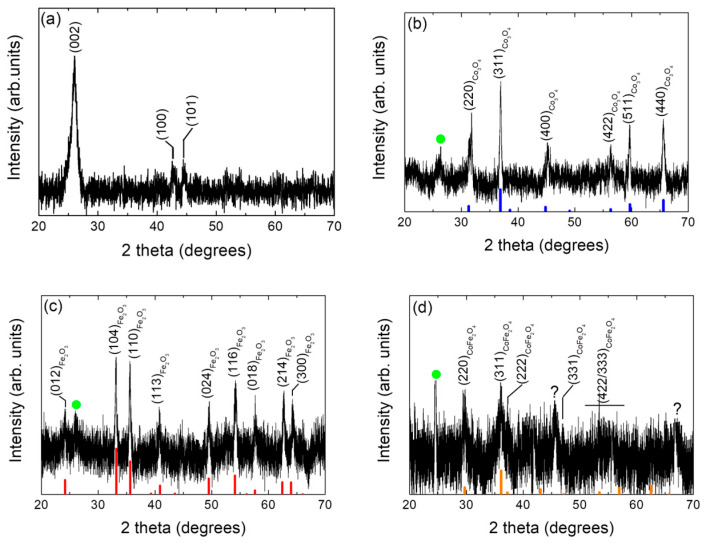
X-ray powder diffraction patterns of the (**a**) single MWCNTs, (**b**) Co_3_O_4_/MWCNTs/IL, (**c**) α-Fe_2_O_3_/MWCNTs/IL, and (**d**) CoFe_2_O_4_/MWCNTs/IL nanocomposite samples. The corresponding crystallographic diffraction planes are indicated through Miller indexes (hkl). The signals marked with a green symbol (●) correspond to the MWCNTs phase. Cubic Co_3_O_4_, hexagonal α-Fe_2_O_3_, and cubic CoFe_2_O_4_ JCPDS patterns are also shown for comparison (Co_3_O_4_, α-Fe_2_O_3_, and CoFe_2_O_4_ JCPDS: vertical thick blue, red, and orange bars, respectively).

**Figure 3 ijms-25-07043-f003:**
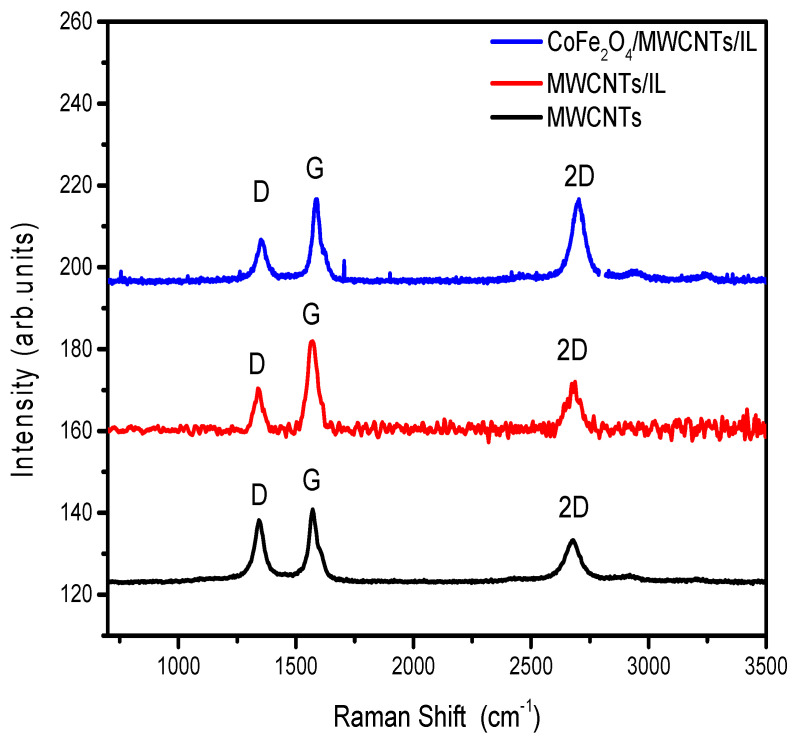
Raman shift of the MWCNTs (black line), MWCNTs/IL (red line), and CoFe_2_O_4_/MWCNTs/IL (blue line) materials.

**Figure 4 ijms-25-07043-f004:**
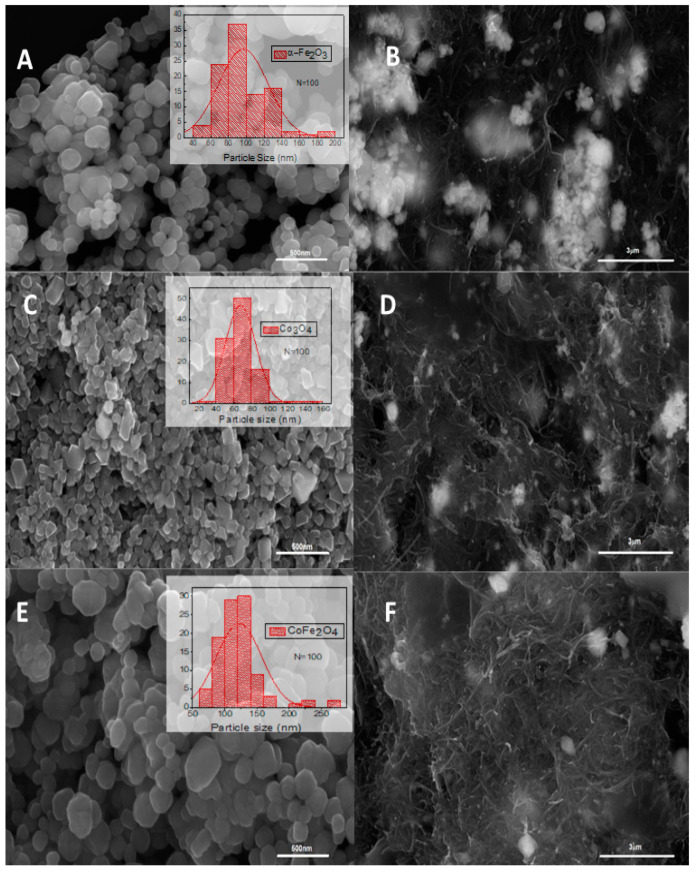
FE-SEM micrograph images of the different studied transition metal oxides (left row) and metal oxides/MWCNTs/IL composites (right row): (**A**) Fe_2_O_3_, (**B**) Fe_2_O_3_/MWCNTs/IL, (**C**) Co_3_O_4_, (**D**) Co_3_O_4_/MWCNTs/IL, (**E**) CoFe_2_O_4_, and (**F**) CoFe_2_O_4_ /MWCNTs/IL. Inset: the corresponding particle size histograms.

**Figure 5 ijms-25-07043-f005:**
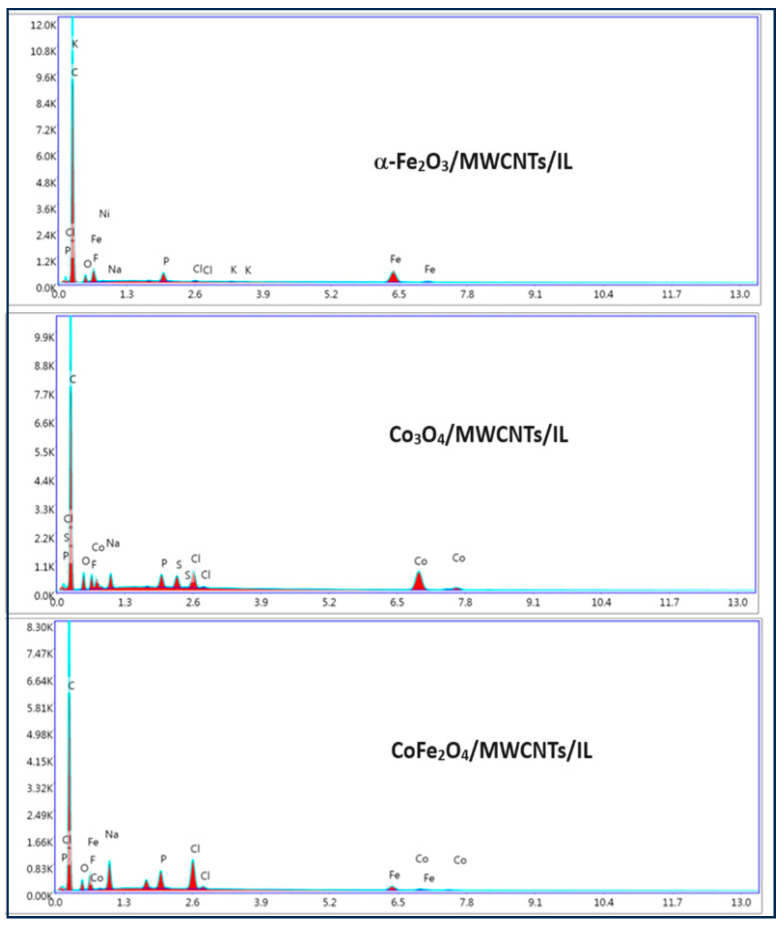
EDS analysis of the different metal oxides/MWCNT/IL nanocomposites: Fe_2_O_3_/MWCNTs/IL, Co_3_O_4_/MWCNTs/IL, and CoFe_2_O_4_/MWCNTs/IL.

**Figure 6 ijms-25-07043-f006:**
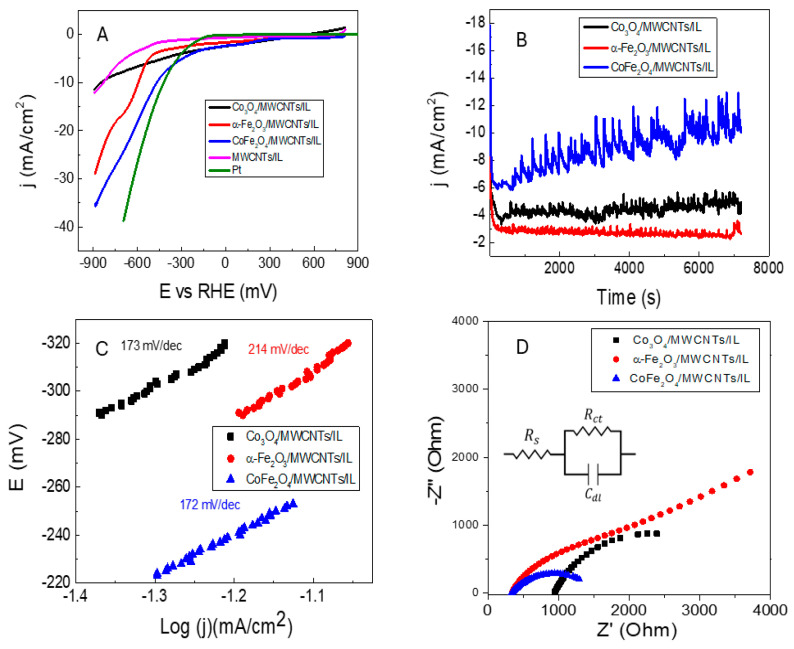
Electrochemical studies of the MWCNTs/IL sample and the different studied transition metal oxides/MWCNTs/IL nanocomposite electrocatalysts as indicated: (**A**) Linear scan voltammetry (LSV) at a scan rate of 100 mVs^−1^, (**B**) HER chronoamperometry stability test at E = −890 mV for the different electrocatalysts, (**C**) Tafel plots for HER and (**D**) Nyquist plots. Inset panel (**D**) Randless equivalent electronic circuit model used to fit the experimental EIS data. Electrochemical studies were carried out in a 0.1 M phosphate buffer electrolytic bath solution, pH = 7.

**Figure 7 ijms-25-07043-f007:**
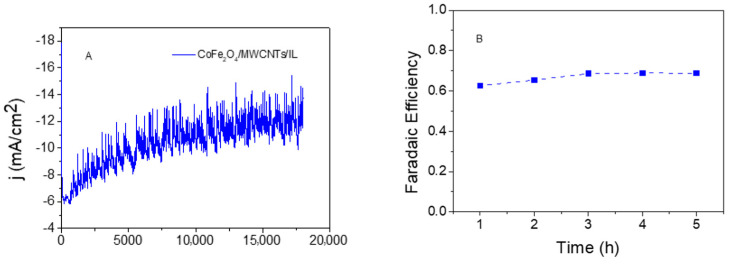
(**A**) Chronoamperometry test profile of five hours of electrolysis for the CoFe_2_O_4_/MWCNTs/IL electrode E = −890 mV. Electrochemical studies were carried out in a 0.1 M phosphate buffer electrolytic bath solution pH = 7. (**B**) Faradaic efficiency at different times of the electrolysis for the composite studied.

**Table 1 ijms-25-07043-t001:** ^1^H-NMR shift (d, ppm) of the organic cation of the ionic liquid N-octylpyridinium hexafluorophosphate.

	H2, H6	H3, H5	H4	2× H1′	2× H2′	2× H3′–H7′	3× H8′
d (ppm)	8.69	8.00	8.46	4.55	1.99	1.27	0.83

**Table 2 ijms-25-07043-t002:** Crystallite size domain of the different synthesized transition metal oxide materials obtained by Scherrer‘s law [30,31].

Material	Crystallite Size (nm)	Crystal Lattice Parameter (Å)
Co_3_O_4_	37 nm	8.059 Å
α−Fe_2_O_3_	27 nm	5.008; 13.63 Å
CoFe_2_O_4_	34 nm	7.947 Å

**Table 3 ijms-25-07043-t003:** Intensities and coefficients of the D, G, and 2D bands of the MWCNTs (black line), MWCNTs/IL (red line), and CoFe_2_O_4_/MWCNTs/IL (blue line) electrode materials.

Material	D	G	2G	I_D_/I_G_	I_2D_/I_G_
MWCNTs	12.89	15.71	11.51	0.82	0.73
MWCNTs/IL	10.62	21.01	9.44	0.505	0.449
CoFe_2_O_4_/MWCNTs/IL	9.091	18.60	18.24	0.48	0.98

**Table 4 ijms-25-07043-t004:** Tafel plot values of metal oxides/MWCNT/LI/nanocomposites at low overpotentials.

Nanocomposite Electrocatalyst	Tafel Slope (mV dec^−1^)
Co_3_O_4_/MWCNTs/IL	173
α–Fe_2_O_3_/MWCNTs/IL	214
CoFe_2_O_4_/MWCNTs/IL	172

**Table 5 ijms-25-07043-t005:** Solution resistance (Rs), charge transfer resistance (Rct), and constant phase element (CPE-P) of the different metal oxides/MWCNTs/LI nanocomposites.

Nanocomposite Electrocatalyst	Rs (Ω)	Rct (Ω)	CPE-P
Co_3_O_4_/MWCNTs/IL	942.2	2715	0.72
α–Fe_2_O_3_/MWCNTs/IL	339.0	1933	0.73
CoFe_2_O_4_ /MWCNTs/IL	334.0	1211	0.56

**Table 6 ijms-25-07043-t006:** Comparison of hydrogen production in different systems studied.

Electrocatalytic Material	Hydrogen Production (μmol cm^2^ h^−1^)	Efficiency %	Reference
MWCNTs/IL	13.82 μmol cm^2^ h^−1^	58%	[13]
CoFe_2_O_4_/MWCNTs/IL	132 μmol cm^2^ h^−1^	68.8%	This work
SWCNHs/Co (II) complex	1.78 μmol cm^2^ h^−1^	100%	[41]
Carbon cloth/CoS	20 μmol cm^2^ h^−1^	100%	[42]
Cu(II) oxime complex	105 μmol cm^2^ h^−1^	100%	[43]
WO_3_	55 μmol cm^2^ h^−1^	-----	[44]
Pt/C (acid media)	2.32 mmol cm^2^ h^−1^	100%	[45]

## Data Availability

Not applicable.

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
