# Peer review of "α-Fe_2_O_3_/, Co_3_O_4_/, and CoFe_2_O_4_/MWCNTs/Ionic Liquid Nanocomposites as High-Performance Electrocatalysts for the Electrocatalytic Hydrogen Evolution Reaction in a Neutral Medium"

_ijms, 2024, doi:10.3390/ijms25137043_

Round 1

Reviewer 1 Report

Comments and Suggestions for Authors

The paper investigates the synthesis and characterization of α-Fe2O3, Co3O4, and CoFe2O4 nanocomposites supported on multi-walled carbon nanotubes (MWCNTs) in an ionic liquid medium for the electrocatalytic hydrogen evolution reaction (HER) in neutral conditions. The authors demonstrate the superior electrocatalytic performance of the synthesized nanocomposites, attributing it to their unique structural and compositional characteristics. Experimental results suggest that these nanocomposites hold promise as efficient HER catalysts in various applications. However, there are some aspects where revisions are recommended:

1.     The literature review provides a good background for the study; however, the authors could further strengthen the discussion by comparing their findings with those reported HER in existing literature for example (10.1021/acs.energyfuels.3c02448). Highlighting the novelty and significance of the proposed nanocomposites relative to previous studies would enhance the impact of the research.

2.     Are there any mechanistic insights or theoretical models proposed to explain the observed enhancements in electrocatalytic activity?

3.     Beyond the HER, are there other potential applications envisaged for the synthesized nanocomposites?

4.     The manuscript briefly mentions the stability and durability of the nanocomposites as electrocatalysts. It would be beneficial for the reader if the author(s) could expand on this aspect by discussing the long-term stability tests or cycling performance of the catalysts under realistic operating conditions.

Author Response

Dear reviewer 1

We have considered every comment made by the referees. The answers for each question/comment are ordered by each observation done (see Referees_responses document).

In the revised manuscript, all the changes are highlighted in gray.

The paper investigates the synthesis and characterization of α-Fe2O3, Co3O4, and CoFe2O4 nanocomposites supported on multi-walled carbon nanotubes (MWCNTs) in an ionic liquid medium for the electrocatalytic hydrogen evolution reaction (HER) in neutral conditions. The authors demonstrate the superior electrocatalytic performance of the synthesized nanocomposites, attributing it to their unique structural and compositional characteristics. Experimental results suggest that these nanocomposites hold promise as efficient HER catalysts in various applications. However, there are some aspects where revisions are recommended:

  1. The literature review provides a good background for the study; however, the authors could further strengthen the discussion by comparing their findings with those reported HER in existing literature for example (10.1021/acs.energyfuels.3c02448). Highlighting the novelty and significance of the proposed nanocomposites relative to previous studies would enhance the impact of the research.

R: We appreciate the reviewer's suggestion to further strengthen the discussion by incorporating a comparison with existing literature. In the revised manuscript, we have expanded the literature review section to include a  comparison of our findings with those reported in relevant studies

  1. Are there any mechanistic insights or theoretical models proposed to explain the observed enhancements in electrocatalytic activity?

R: We acknowledge the reviewer's request for deeper mechanistic insights and theoretical models to explain the observed enhancements in electrocatalytic activity. While the current study primarily focused on the synthesis, characterization, and electrochemical performance evaluation of the catalyst, we recognize the importance of understanding the underlying mechanisms.

The Tafel slope analysis provides some preliminary insights, suggesting that the mechanism involved may have volmer-heyrovsky mechanism. However, a comprehensive mechanistic investigation, including theoretical modeling and additional experimental studies, would be necessary to fully elucidate the reaction pathway and the specific role of each component in the catalyst.

We also think the enhancement of the activity is related to the inherent characteristics of the composite material, especially cobalt ferrite, characteristics that are now deeply explained in the manuscript.

  1. Beyond the HER, are there other potential applications envisaged for the synthesized nanocomposites?

R: We appreciate the reviewer's interest in the broader applicability of the synthesized nanocomposites. While our current study focused on the Hydrogen Evolution Reaction (HER), we believe these materials hold promise for a variety of other applications, especially regarding energy reactions such as Oxygen evolution reaction, Oxygen reduction reaction, Ammonia oxidation, hydrazine sensor, etc.

  1. The manuscript briefly mentions the stability and durability of the nanocomposites as electrocatalysts. It would be beneficial for the reader if the author(s) could expand on this aspect by discussing the long-term stability tests or cycling performance of the catalysts under realistic operating conditions.

R: We acknowledge the reviewer's request for more information on the long-term stability of the nanocomposites. In the manuscript, we have provided evidence of electrochemical stability through a two-hour electrolysis test at a fixed potential (Figure 6B). This test demonstrates that the studied systems maintain their electrocatalytic activity over time, with no observed drop in current density. This finding corroborates the robustness of the nano carbonaceous pastes linked to the nanostructured transition metal oxides, suggesting their suitability for long-term use in industrialized application

Reviewer 2 Report

Comments and Suggestions for Authors

1.      In the introduction section, the article does not mention or discuss why IL causes carbon materials to aggregate or why such composite materials can improve electrochemical activity and conductivity. Need to strengthen the explanation.

2. In the introduction section, the research demonstrates the use of transition metals with ILs to replace precious metals (i.e., Pt, Ir, etc.) in HER applications. However, the cost of transition metal/IL/MWCNTs used compared with those well-known precious metals has not been studied in detail. In Table 6, there is no comparison with precious metals, so it is difficult to convince the value of this work.

3. The article's title mentions neutral electrolytes, which seems to be a highlight of this work. However, the introduction section and the data in the main context do not indicate/support this related discussion.

4. There is a lack of direct evidence for the existence of CoFe2O4 for XRD experiments. The cited literature 22 showed that CoFe2O4 has a (311) peak, and the doping only causes peak broadening and shift. However, the article claimed that doping caused the (311) peak to disappear. Such data is unconvincing.

5. Following the above, why did the broad carbon peak of MWCNTs (the (002) peak that should fall at 25-28) completely disappear? According to the proportion of catalytic ink (served as cathode working electrode) indicated in Methods, the proportion of MWCNTs is very high; as a result, the (002) peak should exist.

6. D, G, and 2D bands in Raman illustrate the information of “molecular” defects and interlayer spacing of the carbon structure. In my opinion, the use of IL should only cause aggregation of carbon materials due to intermolecular van der Waals forces and/or polar attractions between IL and carbon materials. I do not think that adding IL causes fewer defects in the carbon material molecules, so it may not be significant to explore the mutual ratio among D, G, and 2D bands. This may just be associated with the preparation of the sample. The authors must provide more convincing data and highly relevant references to show a stricter discussion and the relevance of Raman's experiments to HER.

7. Following the above, I consider the author wants to indicate the conductivity increase by adding IL. If so, samples should be carefully prepared, and then a measurement of electronic conductivity (i.e., four-point probe) should be conducted.

8. In Figure 6B, please explain why the CA current of CoFe2O4/MWNCT/IL is getting larger.

9. In Figure 6C, all materials exhibited greater values of Tafel slopes over 118 mV/dec. This is not a typical Volmer-Heyrovsky mechanism. There should be other slow steps that cause the rate-determining step to shift. The authors should provide more relevant experiments and verifications to figure out the RDS and make a meaningful explanation.

10. In Figure 6D, the author should explain why the solution impedance (Rs) of the Co3O4 sample is larger than other samples.

11. Please correct the Rtc to Rct in Table 5.

12. In Figure 7, please explain why the CA current is getting larger. Is there chemical evolution and phase change on the catalyst surface? The author must provide ex-situ experiments (EIS, XPS, XRD, etc.) to explain.

13. The authors should provide the corresponding Faradic Efficiencies at different times in the 5-hour period.

14. There are too few comparison objects in Table 6, and the authors should add the comparison with those well-known precious metals.

15. In the conclusion section, the author claimed that “the improved electrochemical activity is probably due to the interaction of both cations (Co, Fe) in the same structure and the interaction with oxygen vacancies.” However, this work did not reveal the existence of oxygen vacancies by any material characterizations. Overall, the data provided in this work are weak in supporting the relevant improved HER performance and the corresponding narratives.

Author Response

Dear reviewer 2

We have considered every comment made by the referees. The answers for each question/comment are ordered by each observation done (see Referees_responses document).

In the revised manuscript, all the changes are highlighted in gray.

  1. In the introduction section, the article does not mention or discuss why IL causes carbon materials to aggregate or why such composite materials can improve electrochemical activity and conductivity. Need to strengthen the explanation.

R: Thank you for raising this point. We acknowledge that the introduction could provide a more in-depth explanation of the mechanisms behind IL-induced aggregation and the resulting improvements in electrochemical activity and conductivity.

To delve deeper into this topic, the following paragraph was added in the introduction: 

Among the different carbonaceous pastes, the pastes formed with carbon nanotubes and agglutinated with ionic liquids improve the activity of different electrochemical reactions due to several synergistic effects. [10] The ionic qualities of the ionic liquids, such as their high ionic conductivity and electrochemical stability, increase the electrode's electrochemical activity and conductivity compared to traditional binders like mineral oil. This enhancement arises from the ionic liquids’ ability to facilitate charge transfer at the electrode-electrolyte interface, improving the kinetics of the electrochemical reactions. Additionally, the ionic liquid's ability to disperse and stabilize carbon nanotubes within the paste leads to a more uniform and conductive electrode surface, further boosting its performance. [10–12] In this context, electrodes formed with multi-walled carbon nanotubes (MWCNTs) agglutinated with the ionic liquid N-octylpyridinium hexafluorophosphate (MWCNT/IL) showed good activity toward hydrogen production. [13] This particular combination leverages the unique properties of both MWCNTs, with their high surface area and electrical conductivity, and the ionic liquid, with its charge transport capabilities. The resulting MWCNT/IL electrode exhibits enhanced electrocatalytic activity for the HER.

  1. In the introduction section, the research demonstrates the use of transition metals with ILs to replace precious metals (i.e., Pt, Ir, etc.) in HER applications. However, the cost of transition metal/IL/MWCNTs used compared with those well-known precious metals has not been studied in detail. In Table 6, there is no comparison with precious metals, so it is difficult to convince the value of this work.

R: We appreciate the reviewer's insightful comment regarding the cost comparison. We acknowledge that a detailed cost analysis comparing the transition metal/IL/MWCNTs system with precious metals was not included in the original manuscript.

To solve that, we add the Pt activity response in the figure 6A to appreciate the comparative activity to the developed composites.

Regarding table 6, those compounds are related to the hydrogen production quantification comparison, and there was not findings of hydrogen production in mol/h cm2 for precious metals in neutral media.

  1. The article's title mentions neutral electrolytes, which seems to be a highlight of this work. However, the introduction section and the data in the main context do not indicate/support this related discussion.

R: We appreciate your feedback regarding the clarity of the title's connection to the neutral electrolyte discussion. While the significance of neutral electrolytes is central to our research, we agree that the introduction and data could better showcase this.

To solve that, the next paragraph was added :

Hydrogen evolution reaction (HER) is vital for sustainable hydrogen production. In this context, electrolysis in neutral media presents a compelling approach with unique advantages. Notably, neutral electrolytes minimize the corrosion of electrodes and enhance material compatibility, addressing challenges in acidic or alkaline environments. This expanded compatibility broadens the range of suitable electrocatalysts, contributing to cost reduction and promoting wider options for green hydrogen production.

Additionally, the neutral media conditions was discussed in the point 9 regarding tafel plots

  1. There is a lack of direct evidence for the existence of CoFe2O4 for XRD experiments. The cited literature 22 showed that CoFe2O4 has a (311) peak, and the doping only causes peak broadening and shift. However, the article claimed that doping caused the (311) peak to disappear. Such data is unconvincing.

R: We agree with the Referee that with the originally presented XRD patterns, it is very difficult to ascertain the presence of the CoFe2O4 phase. Moreover, the discussion and specifically the statement that the ¨doping caused the (311) peak to disappear¨ was not correct.   

To solve that the XRD measurements of the CoFe2O4 samples have been made again, and the new results are depicted in Figure 2d, and then a new discussion about these new results has been introduced. The discussion about the doping effect on the absence of the (311) peak has been deleted. Additionally, the XRD and FT-IR spectra of the metal oxides were added to the supplementary data.

  1. Following the above, why did the broad carbon peak of MWCNTs (the (002) peak that should fall at 25-28) completely disappear? According to the proportion of catalytic ink (served as cathode working electrode) indicated in Methods, the proportion of MWCNTs is very high; as a result, the (002) peak should exist.

R: Yes, with the proportion of the MWCNTs in the electrocatalytic composite, the corresponding XRD diffraction peaks should undoubtedly appear. In the present case, the problem was that the scale used in the plot was very large, and the small peaks could not be seen, which is why it is necessary to zoom in on these plots.

To solve that, now, in the revised version of this MS, the XRD figures have been remake and the ordinate scale has been chosen in order to visualize the presence of the MWCNTs phase diffraction peaks.

  1. D, G, and 2D bands in Raman illustrate the information of “molecular” defects and interlayer spacing of the carbon structure. In my opinion, the use of IL should only cause aggregation of carbon materials due to intermolecular van der Waals forces and/or polar attractions between IL and carbon materials. I do not think that adding IL causes fewer defects in the carbon material molecules, so it may not be significant to explore the mutual ratio among D, G, and 2D bands. This may just be associated with the preparation of the sample. The authors must provide more convincing data and highly relevant references to show a stricter discussion and the relevance of Raman's experiments to HER.

R: While we agree that ILs are unlikely to directly reduce molecular defects, we believe the structural changes they induce, added with the metal oxides, are still highly relevant to the HER performance. Especially with the aggressive method of sample preparation, which includes the use of organic solvents and mechanic homogenization with an agate mortar 

The interlayer spacing and degree of aggregation can significantly influence the  accessibility of active sites, mass transport, and electronic properties of the carbon material, all of which are crucial factors for HER activity.

  1. Following the above, I consider the author wants to indicate the conductivity increase by adding IL. If so, samples should be carefully prepared, and then a measurement of electronic conductivity (i.e., four-point probe) should be conducted.

R: We appreciate the reviewer's suggestion to conduct four-point probe measurements to directly assess the electronic conductivity changes upon IL addition. As noted in the introduction, the enhancement of conductivity by ionic liquids is a well-established phenomenon in the literature. Our study builds upon this established principle and focuses on its application to HER catalysis. While direct conductivity measurements would be valuable complementary data, we believe the existing evidence, combined with the HER performance results, strongly supports the hypothesis that ILs improve the conductivity and overall performance of the catalyst system.

  1. In Figure 6B, please explain why the CA current of CoFe2O4/MWNCT/IL is getting larger.

R: See point 12.

  1. In Figure 6C, all materials exhibited greater values of Tafel slopes over 118 mV/dec. This is not a typical Volmer-Heyrovsky mechanism. There should be other slow steps that cause the rate-determining step to shift. The authors should provide more relevant experiments and verifications to figure out the RDS and make a meaningful explanation.

R: We acknowledge that the non-ideal Tafel slopes indicate a complex HER mechanism. However, we believe that this complexity is a reflection of the nature of our catalyst system and the diverse interactions between the components.

While a complete mechanistic understanding is a long-term goal, the current study focuses on demonstrating the practical feasibility and performance of this novel catalyst for HER. The high activity and stability observed, even with a complex mechanism, highlight the potential of this system for practical hydrogen production applications. Further mechanistic investigations will undoubtedly provide valuable insights for future optimization and design of HER catalysts

For this case, and for the values of tafel slopes obtained, we believe that could be volmer- heyrovsky mechanism, which in neutral media needs an extra overpotential for hydrogen production.

In the manuscript we added that:  The reason for the values over the typically 120 mv/dec could be that in electrocatalytic water splitting for H2 production in neutral or alkaline media, HER kinetics are slowed by the requirement of an additional water dissociation step, making the electrocatalyst need extra energy to gain a proton for H2 evolution.

  1. In Figure 6D, the author should explain why the solution impedance (Rs) of the Co3O4 sample is larger than other samples.

We appreciate the observation made by the Reviewer. However, the explanation of the values ​​of Rs is given in the manuscript according to the sentence: "The high ohmic resistance values ​​of the Rs solution are striking, which would be due to the presence of part of the electrode bound with mineral oil, which electrically insulates these surfaces". This explanation is valid for all samples. In the case of Co3O4/MWCNTs/IL, this sample could have presented a higher degree of oil impregnation, which would explain its higher value of Rs. In fact, this agrees with the resistive profile observed in the linear voltammetry profile performed. This observation is also indicated in the manuscript.

 We change the paragraph in the manuscript for better redaction and explanation :

 “ The elevated solution resistance (Rs) values observed are likely attributable to the consequences of part of the electrode bound with mineral oil, which electrically insulates these surfaces. This is particularly evident in the Co3O4/MWCNTs/IL composite, which added to the inherent low conductivity of the cobalt oxide, presenting a larger Rs value than the other electrodes. This behavior can be confirmed in the slightly resistive profile observed in the linear sweep voltammetry of this composite.”

  1. Please correct the Rtc to Rct in Table 5.

R: Done

  1. In Figure 7, please explain why the CA current is getting larger. Is there chemical evolution and phase change on the catalyst surface? The author must provide ex-situ experiments (EIS, XPS, XRD, etc.) to explain.

R: We appreciate the observation made by the Reviewer. Considering the nature of the material (oxide) and the HER process, a phase change on the catalyst surface is not feasible. However, as observed in the FE-SEM image (see Fig. 4 E and F) it is more probable that the increase in current density shown in Fig. 7 (until reaching a plateau) can be attributed to an increment in the effective area of the catalyst with the immersion time in the solution. It can penetrate the interstices of the material, favoring the increase of the area over the geometric area used to calculate the current density. In fact, the value found by simulation in the EIS response (CPE-P) shown in Table 5 would demonstrate this effect.

To solve that, this phrase was added to the manuscript: A gradual increase in current density up to a plateau around -12 mA/cm2 can be observed. This behavior can be attributed to the increment in the effective area of the CoFe2O4/MWCNTs/IL phase in contact with the electrolytic solution according to immersion time. This fact is in agreement with the CPE-P value obtained by simulation of the EIS responses to CoFe2O4/MWCNTs/IL shown in Table 5.

  1. The authors should provide the corresponding Faradic Efficiencies at different times in the 5-hour period.

R: In the revised manuscript, we have now added figure 7 B detailing the FEs at regular intervals throughout the experiment. This data further supports the stability and performance of our catalyst system over extended operation.

  1. There are too few comparison objects in Table 6, and the authors should add the comparison with those well-known precious metals.

R: See point 2.  It was difficult to find catalysts with their hydrogen production ratio in mol/h cm2. Especially in neutral media.

  1. In the conclusion section, the author claimed that “the improved electrochemical activity is probably due to the interaction of both cations (Co, Fe) in the same structure and the interaction with oxygen vacancies.” However, this work did not reveal the existence of oxygen vacancies by any material characterizations. Overall, the data provided in this work are weak in supporting the relevant improved HER performance and the corresponding narratives.

R: We thank the reviewer for their feedback regarding the conclusion section. We have removed the statement about oxygen vacancies, as it was not directly supported by the data presented in this work.

In the revised manuscript, we have revised the conclusion to focus on the well-established role of ferrites in improving electrochemical activity due to their inherent properties, such as high surface area, electrical conductivity, synergistic effects between Co and Fe, etc. We believe this provides a more accurate explanation for the enhanced HER performance observed in our study

Round 2

Reviewer 2 Report

Comments and Suggestions for Authors

1. Suggest the author can note how the chemistry of the ionic liquid affects the HER catalytic steps, including adsorption, intermediate formation, hydrogenation steps, and desorption. It is recommended that the author add such narrative in the introduction.

2. The author should add the experimentally measured HER Faradaic performance and hydrogen production yield of Pt for comparison.

3. Suggest the author should reinforce the discussion about : how the interlayer spacing and degree of aggregation in carbon materials (by Raman profiles) can significantly influence HER activity in views of the accessibility of active sites, mass transport, and electronic properties and etc. Deeper description and proper literature cites are necessary.

4. According to the FE=68.8% HER for the as-prepared materials in this work, the author should disclose the by-products (FE: 31.2%)?

Author Response

  1. Suggest the author can note how the chemistry of the ionic liquid affects the HER catalytic steps, including adsorption, intermediate formation, hydrogenation steps, and desorption. It is recommended that the author add such narrative in the introduction.

 R: We have revised the introduction to address the comment regarding the role of the ionic liquid (IL) in the hydrogen evolution reaction. The precise mechanism by which N-octyl pyridinium hexafluorophosphate influences the HER still needs to be fully elucidated, and the addition of MWCNTs and Cobalt ferrite made that even more challenging to achieve. However, we finally decided to add to the introduction the properties regarding proton adsorption and hydrophobicity with their respective references.

  1. The author should add the experimentally measured HER Faradaic performance and hydrogen production yield of Pt for comparison.

 R: Due to equipment availability limitations, we could not conduct these experiments ourselves. However, we have added a reference to a study that reports the HER performance of Pt in an acid media, providing a benchmark for comparison with our catalyst's performance. Pt yield performance in neutral media is still difficult to find.

  1. Suggest the author should reinforce the discussion about how the interlayer spacing and degree of aggregation in carbon materials (by Raman profiles) can significantly influence HER activity in views of the accessibility of active sites, mass transport, and electronic properties and etc. Deeper description and proper literature cites are necessary.

 R: We appreciate the suggestion for the reviewer. We believe that this revised discussion provides a more comprehensive analysis of the relationship between the MWCNT structure and the observed HER activity, taking into account not only the overall structural integrity but also the subtle changes in crystallinity of MWCNTs that may contribute to improved charge transfer and catalytic performance. Regarding this, the next paragraph was added to the discussion with their respective reference.

 “As mentioned before, the addition of the ionic liquid and cobalt ferrite does not significantly alter the overall defect and graphitic nature of the MWCNTs; the improvement in crystallinity could facilitate more efficient charge transfer along the MWCNTs, thereby enhancing their electrical conductivity and promoting faster electron kinetics during the HER process.”

  1. According to the FE=68.8% HER for the as-prepared materials in this work, the author should disclose the by-products (FE: 31.2%)?

R: We have revised our discussion to acknowledge the possibility that the oxygen reduction reaction (ORR) contributes to the unaccounted current in our system. As was added to the manuscript, the open cell configuration and aqueous electrolyte could facilitate the diffusion of oxygen generated at the counter electrode into the bulk solution, where it could undergo reduction at the working electrode. We propose to investigate this further in future studies utilizing a well-divided cell setup to isolate the working and counter-electrode compartments